# Pre-Hispanic Foods Oyster Mushroom (*Pleurotus ostreatus*), Nopal (*Opuntia ficus-indica*) and Amaranth (*Amaranthus* sp.) as New Alternative Ingredients for Developing Functional Cookies

**DOI:** 10.3390/jof7110911

**Published:** 2021-10-27

**Authors:** Georgina Uriarte-Frías, Martha M. Hernández-Ortega, Gabriela Gutiérrez-Salmeán, Miriam Magale Santiago-Ortiz, Humberto J. Morris-Quevedo, Marcos Meneses-Mayo

**Affiliations:** 1Centro de Investigación en Ciencias de la Salud (CICSA), Facultad de Ciencias de la Salud, Universidad Anáhuac México, Lomas Anáhuac, Huixquilucan 52786, Estado de Mexico, Mexico; gina_uriarte@hotmail.com (G.U.-F.); martha.hernandezor@anahuac.mx (M.M.H.-O.); gabriela.gutierrez@anahuac.mx (G.G.-S.); miriamsantiagoo@anahuac.mx (M.M.S.-O.); 2Centro de Estudios de Biotecnología Industrial (CEBI), Universidad de Oriente, Ave. Patricio Lumumba s/n, Reparto Jiménez, Santiago de Cuba 90500, Cuba

**Keywords:** amaranth, nopal, oyster mushroom, natural ingredients, functional foods, nutraceuticals, polyphenols, antioxidant, cookies

## Abstract

Oyster mushroom (*Pleurotusostreatus*), nopal (*Opuntia ficus-indica*) and amaranth (*Amaranthus* spp.) are pre-Hispanic foods widely consumed in Mexico. However, there are no standard products developed with these ingredientsas functional cookies. This study evaluated the impact of partial replacement (50%) of whole-wheat flour (WWF) with three formulations of *P. ostreatus*, nopal and amaranth flours (POF, NF and AF, respectively) on the nutritional/antioxidant properties of fortified cookies. The proportion of the flours’ ingredients (WWF:AF:NF:POF) were 100% WWF (traditional cookies), 50:35:10:5 (F1), 50:30:15:5 (F2) and 50:40:5:5 (F3). Proximal composition, phenolic/flavonoid contents, and ABTS^•+^ scavenging activity were determined in flours and cookies.POF, NF and AF possess a high nutritional value comprising polyphenols/flavonoids and a significant antioxidant potential. Total protein, ash and flavonoids were higher in fortified cookies than in controls. Cookies prepared with F2—the highest nopal level—contained 5.29% of dietary fiber and five times higher polyphenol content than control cookies. The ABTS^•+^ scavenging ability was similar in the three enriched cookies (87.73–89.58%), but higher than that in traditional cookies (75.60%). The applicability of POF/NF/AF for replacing up to 50% of WWF in the production of functional cookies was demonstrated without compromising products’ acceptability.This research promotes renewable local bioresouces for a sustainable agri-food chain, especially edible mushrooms.

## 1. Introduction

Non-communicable diseases (NCDs) are the main health and development challenge facing humankind all over the world in the twenty-first century. NCDs, namely cardiovascular diseases, diabetes, cancer and chronic respiratory diseases, have a higher morbidity and mortality rate globally than all other causes combined. By 2030, the global average age-standardized NCDs mortality rate could be 510.54 (per 100,000 population), and the global average mortality for NCDs deaths of the total number of deaths could be 75.26% [1]. On the other hand, malnutrition is one of the main public health problems affecting many countries and is associated, in many cases, with the high intake of hypercaloric foods with low dietary fiber, minerals and vitamins [2].

Given this problem, the search for alternatives is necessary, such as functional foods and nutraceuticals (FF/Ns), which have received considerable interest in the past decade, largely due to increasing consumer awareness of the health benefits associated with food and nutrition [3]. In the current pandemic of NCDs and malnutrition, a central element of the activities to fight against these nutritional diseases and improve the diet of the n should be the maximum possible recovery of the traditional Mexican diet and food heritage [4]. FF/Ns have been consumed since pre-Hispanic times as a part of the regular diet, and some of them have transcended and are consumed in different countries around the globe. Examples of iconic pre-Hispanic FF/Ns in the Mexican and Central American diet include amaranth, nopal and mushrooms (mainly the oyster mushroom *Pleurotus ostreatus*), as well as cacao, corn (maize), peppers (chile), beans, chia and *Spirulina* algae, among others [5]—hence the interest in making culturally rooted foods with greater nutritional value and medical benefits in order to contribute to a healthier diet of the population.

Edible and medicinal mushrooms are now the emerging and most important microbial agri-food chain in Mexico, showing a good impact on food production for human consumption and as an endless source of novel compounds with medicinal properties, which are different from those found in foods of plant and animal origin [6].

*Pleurotus* spp. is the second most important mushroom of culinary value worldwide. *Pleurotus* species have been recognized as a high-nutritional value food containing bioactive molecules with therapeutic effects [7]. In this context, *Pleurotus ostreatus* is highlighted by its therapeutic properties, such as anti-inflammatory, antimicrobial, antiviral, antitumor, antioxidant, antimutagenic, cardioprotective, antidiabetic, and immunomodulatory activities, among others [8,9]. These effects are attributed to the presence of biologically active compounds such as polysaccharides, peptides, proteins, glycoproteins, polyphenols, nucleotides, triterpenoids, lectins, lipids, and other complex compounds [10]. In particular, the antioxidant properties of *P. ostreatus* have been well documented in both mycelium and fruiting bodies at different maturation stages [11,12].

*Pleurotus* mushrooms have become a staple food because of the variety in processing and consumption. In previous studies, some *Pleurotus* species were investigated as food additives: (i) the addition of *P. sajor-caju* to selected wheat- and rice-based products [13]; (ii) the incorporation of oyster mushroom into biscuits in the range of 5–30% [14]; (iii) the production of Fettuccine pasta with partial replacement of wheat flour with *P. ostreatus* flour at 10% and 20% [15]; and (iv) sponge cakes supplemented with dried *P. sajor-caju* powder at 5%, 7%, 10% and 12% [16]. However, the use of *Pleurotus* spp. in the formulation of industrialized food is not fully exploited.

An innovative and unexplored approach consists of combining *Pleurotus* mushrooms (e.g., dehydrated powder) with other traditionally consumed foods such as nopal (*Opuntia ficus-indica* (L.) Mill) and amaranth (*Amaranthus* sp.), taking advantage of their high nutritional value and biological activities.

Nopal is by far the most important cactus worldwide, and the domestication of this plant was one of the most important legacies left by ancient people in Mexico, which is the largest producer [17]. During the last few decades, several scientific studies have been conducted on *Opuntia ficus-indica*, promoting its nutritional value, due to a composition rich in amino acids, polyunsaturated fatty acids, vitamins and dietary fiber contents [18,19]. Additionally, there is ample evidence of the health benefits of nopal consumption, including anticancer properties or cancer chemoprevention issues, and antidiabetic, cardioprotective, anti-inflammatory, antibacterial and antiulcer activities, among others [20]. Recent trends in nutraceuticals research aroused scientists’ interest in studying the effects of nopal bioactive compounds—mainly polyphenols—to scavenge free radicals in oxidative stress-related diseases [21,22].

On the other hand, amaranth is one of the oldest known edible vegetables, found in Tehuacan, Puebla, Mexico, about 4000 BC ago in the earliest archeological records, and was a major food crop of the Aztecs. This pseudocereal is considered a “superfood” because it possesses a high nutritional value, such as a high-quality protein, unsaturated oils, dietary fiber, tocopherols, phenolic compounds, flavonoids, vitamins, and minerals [23]. Many health effects have been attributed to this plant, comprising hypocholesterolemic activity, an influence on the immune system, antitumor effects, an impact on blood glucose levels, effects on liver functions, celiac disease, and antiallergic action [24]. In view of its phytochemicals, amaranth can be considered as an antioxidant rich food [25].

An attempt was made to elaborate formulations for nutritional bars based on the aforementioned foods, and the effects of these formulations were evaluated in a murine model of diet-induced cardio metabolic disruption [26]. However, in the area of food technology, an important market for cookies as very popular snacks has emerged because they are ready-to-eat, quite durable products with long shelf life [27]. Moreover, they represent a matrix suitable for fortification, thus providing an opportunity for the intake of important nutrients [28]. To the best of our knowledge, there are no standard products developed with *Pleurotus* mushroom, nopal and amaranth in the form of combined functional cookies. These foods have in common a high nutritional value with a good myco-/phytochemical profile and natural antioxidants harboring a great antioxidant potential.

The main purpose of this research was to evaluate the impact of the partial replacement (50%) of wheat flour with three formulations of oyster mushroom, nopal and amaranth flours on the nutritional and antioxidant properties of functional cookies. This study intends to contribute to fortifying traditional types of cookies that lack nutraceutical compounds and antioxidant activity by replacing a significant portion of the wheat flour with iconic culturally rooted foods, thus providing reliable insight into these cookies’ functionality.

## 2. Materials and Methods

### 2.1. Chemicals and Samples

Gallic acid, Folin–Ciocalteu’s reagent, hydrochloric acid, sodium acetate trihydrate, glacial acetic acid, sulfuric acid, aluminum chloride, sodium nitrate, sodium carbonate (anhydrous), acetone, ethanol and sodium hydroxide were purchased from Merck (México, S.A, de C.V., Naucalpan, México).

The 2,2-azino-bis-(3-ethylbenzothiazoline-6-sulfonic acid) (ABTS^•+^), epicatechin, potassium persulfate, ethyl ether were purchased from Sigma-Aldrich Chemical Co. (St. Louis, MO, USA). Redistilled water was used for the preparation of solutions. All chemicals were of analytical grade.

All the baking ingredients of customary quality, such as high-grade Whole-wheat flour (WWF), margarine, white sugar, eggs, vanilla extract and salt, were purchased from the local market. All samples in the original packaging were stored at room temperature until further processing.

### 2.2. Mushroom and Plant Material for Biofunctional Flours Preparation

The mushroom and plant material used in this work for biofunctional flour preparation are shown in Figure 1. Fresh *P. ostreatus* (Jacq:Fr.) Kumm (Pleurotaceae) was provided by “Hongos El Dorado GPO” (Ixtlahuaca, Estado de Mexico, Mexico). Mushrooms were sliced into small pieces and dehydrated for 6 h at 57 °C (Gardenmaster food dehydrator, Model FD-1010PC, Nesco^®^ American, Two Rivers, WI, USA). Afterward, the dried material was homogenized for 3 min (homogenizer FOSS 2094, FOSS Centro America, S.A de C.V., Estado de Mexico, Mexico) and then milled (food grinder Foss Tecator 1093, FOSS Centro America, Estado de Mexico, Mexico).

Nopal (*Opuntia ficus-indica* L.) Mill was obtained from the experimental station of the Colegio de Postgraduados (Montecillo, Texcoco, Mexico). Fresh nopal pads were disinfected with Microdyn^®^ (Tavistock Holding AG., produced by Mercancías Salubres, S.A. de C.V., Ciudad de Mexico, Mexico) for 10 min in order to eliminate microorganisms, cut into small sections, and then scalding at 70 °C for one minute. After that, the material was dehydrated for 10 h at 57 °C, homogenized and milled with the equipment previously mentioned. Both *P. ostreatus* and nopal flours (POF and NF, respectively) were packed in a vacuum (Besser Vacum Professional Kitchen equipment FRESH 43, Dignano, Friuli-Venecia Julia, Italy) into airtight closed polyethylene bags and kept in common storage conditions before analysis and application in confectionery.

Amaranth seed flour (AF) was produced in the Plant San Miguel de Proyectos Agropecuarios, S.P.R. de R.S. (Huichapan, Hidalgo, Mexico) and was purchased from Grupo Nutrisol, S.A. de C.V. (Ciudad de Mexico, Mexico). All bags containing biofunctional flours were covered with aluminum foil and stored away from the light.

### 2.3. Formulation and Production of Fortified Cookies with Biofunctional Flours

Standard cookie dough (control) was elaborated according to a traditional method. WWF cookies (control) and biofunctional flour-enriched cookies were prepared based on the different formulations presented in Table 1. Fortified cookies were produced by replacing 50% of WWF with the indicated proportions of biofunctional flours. Within the scope of this study, these cookie formulations were chosen based on the literature reports and the results of preliminary sensory tests for designing a formulation as a nutritional bar containing POF/NF/AF. In the consulted literature, it is not common to substitute more than 50% of WWF, at least when using non-conventional flours, mainly because of significant changes in sensory attributes and consumers’ acceptance [29,30]. The amount of POF to be incorporated in fortified cookies was established at the level of 5% based on a previous work [31] and the health benefits observed at this level in an in vivo intervention study [26]. For all formulations, margarine was creamed, blended with sugar and egg in a mixer, and then WWF and biofunctional flours were sieved into the above dough and mixed again until a homogenous appearance was reached. The dough was rolled, cut into circular pieces and baked at 180 °C for 12 min (convection oven San-Son Combi SHC-10^®^, Naucalpan, Estado de Mexico, Mexico). After that, cookies were cooled at room temperature for 30 min and stored in hermetic containers, at ambient temperature and protected from the light until analysis. The microscopical appearance of cookies was observed at 40× magnification (microscope Olympus SZ4045, Olympus Corp., Tokyo, Japan).

### 2.4. Proximate and Energy Analysis of Flours and Cookies

Proximate compositional analysis of individual flours and cookies was performed in triplicate following the Association of Official Analytical Chemists standard protocols [32]. Chemical composition analyses were carried out for investigating moisture, ash, fat, crude protein and fiber contents in the samples. Total protein was determined by the Kjeldahl procedure (AOAC 976.05, N × 6.25) (Kjeltec 8200, FOSS Centro América, S.A de C.V., Ciudad de Mexico, Mexico); fat by Soxhlet extraction (AOAC 920.39) (Soxtec model 2055, FOSS Centro América, S.A. de C.V., Ciudad de Mexico, Mexico); crude fiber by the gravimetric method (AOAC 962.09) (Velp Scientifica Fiber Raw Extractor, Velp Scientific, Inc., Bohemia, NY, USA); ash by muffle furnace dry ashing at 550 °C for 24 h (AOAC 942.05) (Felisa^®^ model FE 3C1, series 09110120, San Juan de Ocotan, Zapopan, Mexico), and total carbohydrate by difference, subtracting 100 from the sum of the other components.The energy contents of flours and cookies were determined using a Parr calorimeter model 6400 (Parr Instruments Co., Moline, IL, USA), and the results were expressed as kcal/100 g.

Moisture content was determined in a thermogravimetric balance (Moisture Analyzer MB45, OHAUS Corp., Parsippany, NJ, USA) by desiccation of a 1 g sample for 10 min at 100 °C. Water activity (a_w_) was determined by a portable a_w_ meter (ROTRONIC^®^ HygroPalm HP23-AW, ROTRONIC AG, BSD, Zurich, Switzerland).

### 2.5. Determination of Total Phenolics and Flavonoid Contents in Flours and Cookies

One-gram samples of flours and cookies were extracted by occasional shaking with 70:29.5:0.5 mixture of acetone-water-acetic acid (10 mL) at 37 °C for 60 min in the dark. Supernatant was obtained by centrifugation at 4000 rpm for 30 min at 4 °C (Eppendorf 5702, Eppendorf North America Inc., Framingham, MA, USA). The extraction was repeated by adding 5 mL of the solvent mixture, and the supernatants were mixed and collected in individual vials and stored at −20 °C until use.

Total phenolic contents (TPC) of extracts were determined according to the procedure reported by Singleton and Rossi [33] as described by Ramirez-Sánchez et al. [34]. An aliquot of diluted extracts (10 μL) was mixed with Folin–Ciocalteu’s phenol reagent at 10-fold dilution (75 μL) and allowed to react for 5 min in the dark. Then, 75 μL of a sodium carbonate solution at 75 g/L was added, and the mixture was shaken. After reacting for 90 min at room temperature in the dark, absorbance was measured at 765 nm in an Epoch microplate spectrophotometer (Biotek Instruments Inc., Winooski, VT, USA). The results are expressed as micrograms of gallic acid equivalents per gram of the sample, μg (GAE)/g.

Total flavonoid contents (TFC) in extracts were determined according to the procedure of Zhishen et al. [35]. An aliquot of extracts (10 μL) was mixed with 3 μL of a 5% NaNO_2_ solution and allowed to react for 6 min. Then, 10% AlCl_3_ (3 μL) was added and the mixture was left to react for 5 min. Afterwards, 20 μL of 1 mol/L NaOH solution and 10 μL of distilled water were added and absorbance was measured at 510 nm in the aforementioned spectrophotometer. The results were expressed as micrograms of epicatechin equivalents per gram of the sample, μg (ECE)/g.

### 2.6. Evaluation of Antioxidant Activity

The ABTS^•+^ scavenging ability of extracts prepared from flours and cookies was determined by the procedure reported by Re et al. [36]. The test measures the antioxidants’ capacity to neutralize the 2,2′-azino-bis-(3-ethyl-benzothiazolin-6-sulfonic acid) (ABTS^•+^) stable radical cation, a blue-green chromophore of maximum absorption at 734 nm. A stock solution of ABTS^•+^ cation radical was produced by the reaction between 7 mM water solution of ABTS^•+^ and 2.45 mM potassium persulfate (1:1). The obtained solution was stored at room temperature in the dark for 12–16 h before use. The ABTS^•+^ radical solution was diluted with methanol to reach an absorbance of 1.1 ± 0.02 at 734 nm when using a microplate reader Epoch spectrophotometer (Biotek Instruments Inc., Winooski, VT, USA). Ten microliters of each sample was mixed with 190 μL of ABTS^+^ solution and allowed to react for 2 h in the dark.ABTS^•+^ radical scavenging activity was evaluated by measuring the sudden drop in absorbance at 734 nm in the presence of antioxidants. The antioxidant (AO) capacity was expressed as a percentage of the decoloration of the ABTS^•+^ radical:% = (AR − AS)/AR × 100(1)
where AR is the absorbance of the ABTS^•+^ solution and AS is the sample’s absorbance.

### 2.7. Microbiological Analysis

Microbiological analysis was carried out in agreement with the Mexican Official Standards NOM-247-SSA1-2008 [37], in particular the specifications for cereals and their products. The samples (10 g) were randomly selected from each formulation of fortified cookies and aseptically removed using a flame-sterilized microbiological spatula. Each sample was mixed with 90 mL sterile peptone water 0.1% (Merck, Darmstadt, Germany), and serial dilutions were made to obtain a dilution of 10^4^. The dilutions were analyzed for aerobic mesophilic microorganisms (NMX-F-253), and yeasts and molds (NMX-F-255).

### 2.8. Sensory Acceptance Test

Sensory analysis of the cookies was performed using an effective acceptance test with 40 untrained panelists (25 female and 15 male) in the age range of 22–66. The sensory test was conducted one hour after baking trials using a 9-point hedonic scale ((1)—“disliked extremely”, (2)—“disliked very much”, (3)—“disliked moderately”, (4)—“disliked slightly”, (5)—“neither liked nor disliked”, (6)—“liked slightly”, (7)—“liked moderately”, (8)—“liked very much”, (9)—“liked extremely”), according to Stone and Sidel [38]. The participants were asked to assess the following attributes: liking of color, liking of aroma, liking of texture, liking of taste, and overall acceptability. The threshold of acceptability was set at 5. The samples were kept at room temperature before testing and were identified with random codes of three numbers.

### 2.9. Statistical Analysis

Experimental data of proximal composition, energy value, TPC and TF, and antioxidant activity were expressed as the mean ± standard deviation (SD) of three replicates from three independent experiments. One-way analysis of variance (ANOVA) followed by the Tukey HSD test was used to determine the significance of differences between formulations at *p* < 0.05. Data of sensory studies were analyzed according to a Kruskal–Wallis test followed by Dunn’s multiple comparison test at *p* < 0.05. All statistical analyses were performed with SPSS version 22.0 (SPSS Inc., Chicago, IL, USA) software package for Windows.

## 3. Results

### 3.1. Production and Analysis of Biofunctional Flours

All-natural oyster mushroom, nopal and amaranth flours were produced by dehydration followed by grinding to the desired particle size of 0.71 mm as the average value determined by sieving, considered as a coarse fraction (>0.15 mm) [39]. Chemical composition of whole wheat flour (WWF) and biofunctional flours is presented in Table 2. The wheat and alternative flours used for cookie production differed significantly (*p* < 0.05) in the content of basic proximate components. Our results showed that POF, NF and AF were rich in crude protein, crude fiber and ash, but lower in carbohydrates than WWF. It is noteworthy that POF, NF and AF had approximately 1.46-, 1.16- and 1.11-fold higher levels of protein and 5.21-, 3.49- and 1.6-fold higher levels of fiber than those of WWF, respectively. The higher ash content in biofunctional flours implied that they contained relatively higher mineral content, in the order NF > POF > AF, with values 12-, 5.14- and 1.96-fold higher than WWF. AF had the highest lipid and carbohydrate content (7.26% and 65.34%, respectively) and consequently, the maximum energy value; in contrast, the lowest energy value was recorded for NF. The moisture content of POF, NF and AF was lower than that of WWF (*p* < 0.05), thus indicating a positive impact in terms of conservation.

In addition to the high nutritional value, POF, NF and AF also possess a significant polyphenol content, decreasing in the order NF > AF > POF, with values 8.6, 1.47 and 1.16 times higher than WWF (*p* < 0.05). Flavonoids were especially higher in NF (64.72 ± 2.70 μg ECE/g) (*p* < 0.05), and lower values were obtained for the rest of the samples. NF and POF exerted scavenging activity against ABTS^•+^ radical, about 1.1 times higher than the control.

### 3.2. Production of Fortified Cookies with POF/NF/AF

In order to demonstrate POF, NF and AF’s applicability in confectionery products’ fortification, an attempt was made to add more value to standard cookies, by replacing up to 50% of WWF. Figure 2 shows representative photographs of doughs (a), obtained cookies (b), and microscopical appearance of cookies at 40× magnification (c). As expected, results on the physical characteristics of fortified cookies samples at all formulations showed similar weights (4.3 ± 0.2 g), determined using an analytical electronic balance, without significant differences from the control cookies (*p* > 0.05).

### 3.3. Functionality of Fortified Cookies

The level of enrichment of the cookies, i.e., functional characteristics of fortified cookies, was determined based on an increase in nutrient and myco/phytochemical content and antioxidant activity. Significant improvement of conventional cookies functionality through the incorporation of POF/NF/AF was confirmed.

#### 3.3.1. Nutritional Content

The results obtained for proximal composition of the cookies prepared from different blends of wheat and alternatives flours are depicted in Table 3. An increase was observed in moisture content among the fortified cookies compared to control cookies (*p* < 0.05). However, all the samples were shown to have a_w_ values of less than 0.5, which means that all cookie formulations had a low percentage of free water for microbial proliferation, leading to a highly stable product.

Fortified cookies had approximately 1.2-fold higher levels of protein and between 1.58- and 2.1-fold higher levels of ashes than those of the wheat flour (especially in F2 and F1) (*p* < 0.05). When biofunctional flours were incorporated into cookies, a reduction in total carbohydrates was observed compared to the control. Cookies prepared with F2 contained 5.29% of dietary fiber, 2.1-fold higher than the control (*p* < 0.05). The result of the lipid content showed no significant differences among fortified cookies; only formulation two had a relatively slight decrease compared to control cookies (*p* > 0.05). The energy values (542–590 kcal/100 g) did not show significant differences among cookie samples (*p* > 0.05). The lowest energy value was recorded for WWF control cookies (*p* < 0.05).

#### 3.3.2. Content of Dietary Polyphenols and Flavonoids

Table 3 shows the TPC and TFC of the fortified cookies. Those made with 100% whole wheat flour had the lowest TPC and TFC of all samples. As expected, POF/NF/AF substituted for WWF enhanced the TPC and TFC significantly in the fortified cookie samples (*p* < 0.05). It is noteworthy that F2, F1 and F3 had approximately 5.02-, 3.81- and 2.49-fold higher levels of polyphenols than those of control cookies. No significant differences were found in TFC between biofunctional cookies, but the values were greater than traditional cookies (*p* < 0.05), ranging from 1.77- to 2.59-times higher in F3 and F2, respectively. Particularly, cookies prepared with F2 possessed the highest TPC and TFC, in agreement with the added nopal level.

#### 3.3.3. Antioxidant Activity

In this study, the radical scavenging activity towards artificial radical ABTS^•+^ species was explored to evaluate the antioxidant properties of cookie samples. The application of this method is widespread due to its numerous modifications, and it can be applied in antioxidant activity determination in both lipophilic and hydrophilic antioxidants, including food samples. Therefore, it is a suitable technique to clarify the impact of POF/NF/AF on the potential functional properties of fortified cookies. The antioxidant activity of different cookie samples is shown in Table 3. The ABTS^•+^ scavenging ability was similar in the three enriched cookie formulations (87.73–89.58%), but on average 1.2 times higher than traditional cookies (75.60%) (*p* < 0.05).

### 3.4. Microbiological Analysis

Microbiological analysis showed that all cookie formulations were within the permissible limits, according to the Mexican legislation (Table 4). These microorganisms are indicators of management conditions or efficiency of the food preparation. They warn of inadequate handling or contamination that increases the risk of the presence of pathogenic microorganisms in the product on a timely basis [39].

### 3.5. Sensory Characteristics of Cookies

The sensory scores of WWF cookies and those produced with different formulations of alternative flours from functional ingredients were evaluated by using nine-pointhedonic scales, and the results are depicted in Table 5. No statistically significant differences in scores for all sensory attributes between the control WWF cookies and enriched cookies were noticeable (*p* > 0.05). In general, F2, with the highest score rating for all sensorial attributes, had a positive impact on the organoleptic properties of cookies. Even though there were no statistical differences among the sensorial attributes of the cookies, panelists most frequently mentioned that cookies enriched with the alternative flours had an acidic aftertaste and enhanced flavor, which improved their overall acceptance.

## 4. Discussion

The development of new fortified flour products capable of influencing metabolism and other health-related conditions is a demanding task. One of the categories of functional additions is components with antioxidative properties, which can reduce the level of oxidative stress in cells [40]. In this context, the effect of different amounts (5–95%) of food components as well as agro-industrial by-products such as banana flour [41], rapeseed press cake [42], soybean meal [43], apple pomace flour [28], spices and herbs [44], prickly peel fruits [45], mushrooms [29], among others, has been analyzed recently on the antioxidant capacity (AO) of biscuits and cookies.

Most of the time, cookies and biscuits are prepared using refined wheat flour, but composite flour is healthier because it improves the nutritional value of bakery products when blended with other types of flour [41]. In this sense, the development of cookies enriched with POF, NF and AF is suitable due to the nutritional and health-promoting properties of these pre-Hispanic Mexican food products. Thus, fortified cookies with POF/NF/AF offer the possibility of introducing substances including antioxidant molecules with beneficial properties for health through the diet.

The use of POF, NF and AF as natural raw materials to enhance the functional properties of bakery products is not a very common practice. Its incorporation into bakery products has been reported individually for the partial replacement of wheat flour, e.g., oyster mushroom *Pleurotus* at levels of 4–30% (suggested percentage of powder substitution of 5–8% in terms of flour basis) [14,29]; for nopal flour at levels of 4–15% [17,46,47]; and for amaranth at levels of 20–100% [48,49]. To our knowledge, this is the first study to report the development of novel wheat and POF/NF/AF blend cookies with improved nutritional and functional attributes with respect to control cookies. The levels in which the biofunctional flours were incorporated into doughs are in agreement with the studies previously mentioned.

The information about the proximate composition and energy is of great interest for biofunctional ingredients to be used in the formulation of health foods and Ns. Information about the complete characterization of flours will allow researchers not only to establish possible combinations between them with the aim to enhance the nutritional profile of the bakery products, but also to be used in other types of products, such as beverages, soups, sauces, or food adjuvants [50]. The WWF and biofunctional flours used for cookie production differed significantly in the content of basic proximate components, and POF, NF and AF showed a better nutritional profile compared to commercial WWF. They are all rich in proteins, fibers, and ash contents, in contrast to low fat levels (excepting amaranth), which make them suitable to incorporate into low-caloric confectionery products. Even if there was variability on the chemical composition values of the selected flours with those obtained by other researchers, the proximate biochemical analysis of POF, NF and AF is in agreement with the composition presented in previous reports (see below).

The *P. ostreatus* composition ranges between 17% and 30.4% for crude protein, 37% and 85% for carbohydrates, 5.3% and 24% for total dietary fiber, 1.6% and 5.0% for fat and 5.9% and 9.8% for ash [29,51,52] depending on the strains, method of cultivation, substrates and growth conditions, etc. On the other hand, nopal chemical composition is not absolute and may vary according to variety, soil factors, cultivation season, and plant age, in intervals between 7.07% and 19.0% for crude protein, 38% and 61.4% for carbohydrates, 5.97% and 55.05% for total dietary fiber, 0.1% and 2.16% for fat and 14.4% and 23.05% for ash [17,22,47]. In addition, the values obtained for AF composition are consistent with those reported in the bibliography, in ranges between 12.4% and 21% for crude protein, 48% and 69% for carbohydrates, 3% and 20.6% for total dietary fiber, 3.24% and 8.6% for fat and 2.6% and 3.8% for ash, also varying with varieties and several ecological factors [48,49,50,53,54].

Additionally, our results demonstrate that POF, NF and AF are vehicles for the delivery of bioactive substances including antioxidant molecules, such as polyphenols/flavonoids. These molecules are considered to be of high scientific and therapeutic interest, because they help to prevent degenerative diseases, cardiovascular diseases and cancers, among others [55]. The content of phenolics is commonly determined by the Folin–Ciocalteu reaction, based on reducing the Folin–Ciocalteu reagent with phenolic compounds in an alkaline state. Although the exact chemical nature of this reagent is not clearly defined, it contains a complex of phosphomolybdic/phosphotungstic acids, which are reduced to obtain a blue chromophore with the maximum absorption at 765 nm [56]. However, the comparison of our results with data regarding the total phenolic/flavonoid content in the literature is difficult because different extraction solvents and substances for calibration are used, and data are variously expressed in quercetin (Q), catechin (C), gallic (GA), or other compound equivalents (E). Moreover, different methods have been used for measuring antioxidant activity based on diverse chemistry principles [56]. TPC was higher in the three biofunctional flours compared to WWF, but a significant TFC was shown only in NF. Although different studies reported that the antioxidant capacity is strongly related to the phenolic profile in diverse food products [50,57], in our study, no significant correlation was found between TPC and AO (r^2^ = 0.518, *p* > 0.05).

Despite the great extent of variability regarding TPC, FC and AO of the examined flours (and/or their sources) described in the literature, our results in general are in line with other reports. The TPC of POF tested in the present work was 599.3 ± 97.42 μg/g, comparatively higher than the values reported in five wild culinary-medicinal species of *Pleurotus* collected from Northwest India (67.6 to 169.2 μg/g) [58]. In another study, the content of total phenols was determined in five extracts of *P. ostreatus* powder obtained with solvents of different polarity. The content of polyphenols in extracts decreased in the order: water > ethanol > acetone > ethyl acetate > n-hexane [59]. TPC of POF was lower than those reported for water and ethanol (1384 and 863.7 μg/g, respectively) but higher than values reported in less-polar solvents. These findings are consistent with the solvent mixture used for extracting polyphenols in our study, acetone-water-acetic acid (70:29.5:0.5). Lower values were obtained for FC in POF compared to a previous report, in which 5.46 and 3.58 mg CE/100 g were reported for fruiting bodies and primordia of *P. ostreatus*, respectively [12]. In this study, ABTS^•+^ radical scavenging activity of POF (93.48%) was higher than preparations from *P. ostreatus* fruiting bodies and mycelium previously investigated with values of 80% and 55%, respectively [10].

On the other hand, all plant parts of nopal (flowers, pulp, seeds, skin, and leaves) are rich in phenols: flavols, flavones, phenolic acids, etc. [19]. The TPC of NF tested in the present work was 4452 ± 46.50 μg/g, comparatively higher than the values between 562 and 905 μg GAE/g and 570 and 2300 μg GAE/g reported in nopal flour and dehydrated nopal, respectively [60], and in the range of 1240–5410 μg GAE/g [61]. However, lower values were obtained for TPC in NF with respect to the by-products of nopal cladodes (2.7–3.7 g GAE/100 g) [62] and in cladodes after convective drying at 45 °C (40.97 g of phenols/kg of samples) [63]. In this study, NF had a low FC (64.72 ± 2.70 μg ECE/g) compared to other reports [61,63,64]. When compared to fresh cladodes, nopal powders/flours had lower flavonoid content, which is probably related to the effect of temperature/time of drying of the material on the degradation of chemical compounds such as flavonoids [65] or collection periods because of the response of the cultivars to the flavonoids was more intense in the rainy period [61].The antioxidant capacity of cladodes has been assessed both in vitro and in vivo. Differences reported might have arisen from different methodologies and *Opuntia* samples used in different studies. In our study, NF exerted prominent scavenging activity against ABTS^•+^ radical (96.30%). Dehydrated nopal cladode by-products showed ABTS^•+^ radical scavenging activity with values of 6.11 and between 52 and 57 μmol Trolox E/g, respectively [60,62].

Several studies have focused on polyphenols in amaranth species, and have resulted in the identification of many phenolic acids and flavonoids. The TPC in AF found in this work (759 ± 17.71 μg GAE/g) is above the values reported in amaranth seeds from *A. hypochondriacus* extracted with methanol, ethanol, and hexane by means of Soxhlet and magnetic stirring, which varied between 160 and 250 μg GAE/g [66] and higher than a commercial AF (<500 μg GAE/g) [50]. The abundance of flavonoids in amaranth has been confirmed and is highly affected by methodology, especially extraction solvent. Extraction with hot water obtained much higher yields of total phenolic compounds (4.23 ± 1.00 mg GAE/g), but the concentration of flavonoids decreased remarkably to 0.08 ± 0.00 mg CE/g [67]. The predominance of rutin and very low levels of nonglycosylated flavonoids among the phenolic compounds is responsible for the low antioxidant potential of amaranth seeds [53]. A significant antioxidant activity was described for a methanolic extract of *A. caudatus* in all the studied in vitro antioxidant models; however, it was extremely effective in scavenging ABTS^•+^ radical activity (IC_50_ 48.75 ± 1.1 μg/mL) [68]. In our study, although the TPC was reasonably high, lower antioxidant activity, at an average of 77.52% ± 1.74%, was measured. These findings are consistent with a previous report, in which amaranth showed a lower antioxidant activity (average 3.26 mmol TE/kg) compared to quinoa, whereas its TPC was higher. Thus, no significant correlation between phenolics and antioxidant potential was observed [69]. Contradictory to this, other studies showed that in vitro antioxidant capacity correlated well with the amount of phenolics in cereals, pseudocereals and legumes [50,57].

Biscuits/cookies are considered as a type of confectionary with a low moisture content, and they can serve as a vehicle for important nutrients if made readily available to the population [70]. Moisture content and water activity are crucial parameters to predict both the stability and safety of the product, with great impact in conservation [71]. In terms of moisture, there are significant differences in content between biofunctional flours (3.34% in AF to 8.36% in POF), showing lower values than WWF, in line with the information in previous reports: 7.04–10.6% for POF [14,29], 4.70–13.33% for NF [47,60,64] and 4–9.07% for AF [48,50,54]. Moisture content is an index of storage of the flours. These low moisture contents of flours could be due to the efficiency of the drying methods used. Indeed, it is well established that high moisture levels above 12% in food products promote susceptibility to microbial growth and enzyme activity, which accelerates spoilage [72]. The moisture content of the composite cookies (3.16–3.32%) was statistically higher than WWF cookies (2.65%), but within the recommended range of 0–10% for the storage of biscuits [45], and the a_w_ values were lower than 0.5. This is advantageous because a reduction in the moisture content of baked products will reduce the proliferation of spoilage organisms, especially molds, thus improving the shelf stability of the product. The present results compare favorably with other studies on WWF cookies fortified with the individual flours, e.g., oyster mushroom (3.97–9.63% at inclusion levels from 5% to 30%) [29], nopal (3.48% at a 4% level) [45], and amaranth, which showed the highest humidity of 12.8% at 24% inclusion [49].

Collectively, the differences in nutritional composition of cookies samples in this study are related to the original chemical compositions of wheat and POF, NF and AF, along with their relative inclusion level in the recipe. In this recipe, 61.4% of the ingredients correspond to margarine and sugar; therefore, the content of nutrients in composite cookies is relatively lower than in POF, NF and AF. As sugar and wheat flour are added in cookie preparation, the carbohydrate content in cookies should be higher than in flours. However, the occurrence of the Maillard reaction in the baking process decreased the carbohydrate content in cookies compared to flours. The products of the Maillard reaction are able to affect the antioxidant capacity of bakery foods [73]. The Maillard reaction products formed during the production of cookies could also act as antioxidants and scavenge free radicals [74], which consequently contribute to better antioxidant activity of the cookies, as confirmed in this study.

The replacement of the 50% of WWF by different levels of POF, NF and AF in fortified cookies resulted in an increase in protein, ash and dietary fiber (in F2), compared to the control. In other studies, protein content ranged from 11.07% to 15.55% in biscuits fortified with 5% up to 30% of the oyster mushroom [14,75], from 8.47% to 10.26% of cookies enriched with 1–5% of mushroom (*Cordyceps militaris*) flour [76], and 18.7% in cookies enriched with 24% of amaranth flour [49]. Similar results were also reported in the protein content of cookies with the addition of sclerotium flour of the edible mushroom *Pleurotus tuber-regium*, which was significantly higher than that of the control cookies [77]. In contrast, relatively slight but not significant changes in the protein content of cookies was observed when wheat flour was substituted with cladode powder at 2.5% to 7.5% (9.08–9.20% in supplemented cookies vs. 9.26% in control) [64]. In our work, the three formulations of fortified cookies contained more protein than WWF cookies, which may be a reason for predicting a reduced glycemic response of the POF/NF/AF enriched cookies. A reduction in carbohydrates was observed infortified cookies. It has been observed that even small amounts of protein in food products were enough to alter the starch digestibility and other functional properties, thus limiting starch degradation and sugar liberation [73]. Moreover, it is widely known that WFF cookies, commonly available on the market, lack good quality protein because of their deficiency in lysine. For this reason, the production of cookies enriched with POF/NF/AF can increase not only the protein content, but also improves the amino acid balance of the final product, due to the contribution of lysine and other essential amino acids by these biofunctional flours [50,78,79].

Substitution of 50% WWF by AF:NF:POF (30:15:5) in F2 had a positive impact in the dietary fiber content of fortified cookies (5.29% vs. 2.58% in control). The importance of dietary fiber in food is well known; its consumption has shown benefits throughout the digestive process. Different physiological and prebiotic effects at the colon level make this nutrient a key component of a healthy diet [80]. Dietary fiber in cookies based on F2 represents 17–21% of the DRV/100 g for women and 15% of the DRV/100 g for men in the group of 31–50 years old vs. 8.6–10% of the DRV/100 g for women and 7% of the DRV/100 g for men in conventional cookies [80]. However, enriched cookies do not meet the requirements for higher fiber foods—a content ≥ 2.5 g per one serving (30 g for cookies) according to the nutrient specifications for Mexican people [81]. In addition, the content of dietary fiber in F2 may also contribute to the overall antioxidant activity of cookies. The substitution of 7%, 9% and 11% of wheat flour with Fibrex^®^ (commercial formulation of sugar beet dietary fiber, Nordic Sugar, Malmö, Sweden) in the formulation of cookies upgraded the antioxidant activity in the DPPH^•+^ radical scavenging activity test [74]. A synergistic effect of dietary fiber and antioxidant activity, possibly through carrier bioactive compounds, was reported by incorporating mushroom bioactive compound in bread [73]. In other studies, the incorporation of 5% prickly pear peel powder into wheat flour improved the total fiber content of the biscuits from 5.9 to 8.1 g/100 g [82], and the dietary fiber levels of biscuits containing oyster mushroom at 5–30% ranged from 2.10% to 2.93% [14,29]. A significant increase in dietary fiber content (0.95–2.42%) was observed in cookies made with cladode powder at 2.5% to 7.5% compared to the control product (0.40%) [64]. Substitution of wheat flour by amaranth in bread has had similar results, increased insoluble dietary fiber and ash contents [83].

Ash content represents all the inorganic minerals contained in the sample, which have important biological roles in the organism. Several reports have documented the abundance of minerals in oyster mushroom, nopal and amaranth, such as Ca, K, Mg, Fe, and those with antioxidant functions—Zn, Cu, Mn and Se—which seem to play a protective role in diseases related to oxidative stress [18,52,54,79]. The higher ash content of the supplemented cookies indicated that they contained a higher mineral content than the control wheat cookies. These changes were due to the elevated levels of these nutrients, especially within the prickly pear cladodes and the oyster mushroom. The results shown here are comparable to previous reports in cookies made with cladode powder (1.80–3.21%), and higher than the control product (1.19%) [64] and biscuits supplemented with oyster mushroom (1.52–3.58%, level of supplementation 5–30%) [29]. Our results were also higher than ash content (0.51–0.62%) in cookies enriched with 1–5% of mushroom (*C. militaris*) flour [76].

The incorporation of POF/NF/AF into wheat flour significantly improved the TPC and AO capacity of composite cookies compared to the control. In addition, phenolics retain their AO activity after the baking process, which has potential health benefits for consumers, although some authors informed that baking could reduce the TPC in the obtained cookies [44]. TPC was particularly high in F1 and F2, which contained the highest NF levels. The antioxidant effect of supplemented cookies was evaluated by ABTS^•+^ radical-scavenging activity. Interestingly, enriched cookies exhibited similar AO activity upgraded 17.5% compared to cookies prepared with 100% WWF, which displayed the lowest ABTS^•+^ radical scavenging activity. Therefore, no direct correlation was found between the TPC and AO activity of fortified cookies. A relatively slight but not significant decrease was observed in AO activity of F3, which contains the highest level of AF—the flour that exhibited the lowest AO activity. It is worth noting that POF, NF and AF are complex matrixes, and the AO may reflect the interactions (synergy or antagonism) among its constituents in addition to phenolics. This AO array contains vitamins and minerals, and for the oyster mushroom dietary fiber, mainly β-glucans-, polysaccharides, polysaccharide–protein complexes [11,12]; and for nopal and amaranth- dietary fiber, betalains, carotenoids, phytosterols, and chlorophylls, among other molecules [19,22].

In other studies, the TPC of biscuits fortified with 4% of prickly pear peel flour (PPF) was 19.1 mg/g, with significant differences between control and composite biscuits. However, the addition of PPF significantly reduced (*p* < 0.05) the TPC of biscuits. The decrease in TPC was attributed to the fact that baked products have drastically reduced levels of phenolic compounds because of the polymerization of polyphenols and the decarboxylation of phenolic acids that occur during thermal treatment. An improvement in the overall antioxidant activity (DPPH^•^ and ferric-reducing power assay -FRAP- values) was observed with the incorporation of PPF into biscuits [45]. On the other hand, at 7.5% level of cladode supplementation, cookies exhibited the highest DPPH^•^ radical-scavenging activity (53.3% on day 1 and 49.43% on day 30), and a comparable trend was also noticed for FRAP [64]. With respect to mushrooms, despite the methods conducted (DPPH^•^ and ABTS^•+^ radical scavenging activity, FRAP and reducing power), the addition of *C. militaris* mushroom flour as a source of natural phenolic compounds remarkably increases the antioxidant activities of cookies [76]. In the case of AF cookies, the DPPH^•^ radical scavenging activity of germinated amaranth cookies was 1.3 and 1.1 times higher than WWF and raw amaranth flour cookies, respectively [48].

The differences observed in the studies could be due to the different structure of the phenolic compounds that might have influenced the AO activities. The AO activity of the compound structure was reported to be dependent on the number and distribution (orthoand para positions) of the active group (OH) [56]. Electron donating groups, especially alkyl and alkoxy groups, were informed to stabilize the phenoxyl radical, leading to the enhancement of the radical scavenging activity. Moreover, the presence of hydrophilic substituents such as sugars that promote the hydroxyl hydration in aqueous media, along with the molecular weight also influence the AO activity of phenolic compounds [84]. Therefore, determination of the phenolic profile of cookie samples and correlational analysis between phenolic compounds and antioxidant activities are worthwhile to be studied further.

Polyphenolic compounds in the added biofunctional flours could also act as non-proteinaceous amylase inhibitors, and therefore, the fortified cookies could be promissory for glycemic index decrease, in agreement with the manipulation of the predictive glycemic response associated with phenolic properties of bread by incorporating mushroom bioactive compounds [73].

The POF/NF/AF enriched cookies contained a relatively high content of fat (33.13% in F2 to 34.10% in WWF on dry matter basis). Consequently, they are at higher risk of quality degradation in terms of oxidative changes. The addition of natural ingredients bearing antioxidant properties or synthetic antioxidants can extend their shelf life by retarding or inhibiting oxidation reactions. However, synthetic antioxidants such as butylated hydroxyanisole (BHA) and butylated hydroxytoluene (BHT) have been reported as controversial with respect to their safety for utilization in food products [85]. Taken together, our results indicate that supplementation of WFF with POF, NF and AF in fortified cookies not only reinforces the nutritional quality, but also contributes to their stabilization against oxidative damage. Overall, enriched cookies have more functional components and effective antioxidant capacity than wheat cookies. Their supplementation could provide the consumers with a novel product with health-promoting benefits.

Sensory testing and consumer-oriented application optimize the likelihood of obtained a positive effect in the food industry for producers, processors and consumers. This is particularly important when designing any new or modified food product [86]. In several studies, bakery and confectionery products with various shares of POF, NF or AF incorporated were assessed for their sensory properties and consumer acceptance. In accordance with previously published studies, cookies produced within the scope of this work with the levels of POF/NF/AF included in the enriched formulations, especially F2, were best rated for all sensory attributes. They were distinguished by typical appearance, corresponding crunchiness texture, and distinctive aroma. This indicated that appropriate levels of functional flours incorporated into cookies increased the acceptance of cookies.

With respect to the use of the single functional ingredients selected in our study for obtaining composite bakery products, the level of substitution of WWF was about 5–40%, with the exception of a report in which AF replaced 20–100% of WWF. However, in terms of taste, the sensory score significantly decreased after 60% addition of AF, which may be due to bitter aftertaste of AF. The overall acceptability score indicated that the cookies prepared with up to 60% AF had most acceptable sensory attributes [30]. This contradicted the result reported by Sindhuja et al. [87], which revealed that the cookies containing 25% AF was found to be most acceptable by the panelists. According to sensorial analysis results, the 4–10% mushroom powder in bakery products showed the best sensory properties [29]. For mushroom (*C. militaris*) flour, the levels of substitution ranges between 1% and 5%, and the cookies prepared by substituting 3% of wheat flour scored the maximum overall acceptability scores, as well as the highest ratings in terms of odor, taste and texture [76]. On the other hand, the evaluation of quality and antioxidant properties of durum wheat breads fortified with 5–10–15% cladodes showed that it is possible to obtain good sensory results until 10% of fortification. Score decay for all samples was noticed at 15% of substitution [88]. This result is contradictory with our findings, which revealed that cookies containing 15% of NF were the best scored by the panelists.

In general, in the present study, the acceptance of the cookies formulations was above 80%, corroborating that the use of POF, NF and AF did not negatively interfere in the sensory properties of cookies. Considering the aforementioned contribution of POF, NF and AF in fortified cookies to their stabilization against oxidative damage, in conjunction with sensory evaluation, the results presented here support the hypothesis that fortified cookies might demand comparatively simple packaging [89]. Innovation in products with high cultural value could be an important factor for greater consumer acceptance. Studies in this regard are necessary in the development of FF/Ns appealing to different populations.

Further studies are needed to evaluate the storability of novel cookies by potential consumers. Additionally, in vivo intervention studies in animal models and humans will be advisable to verify the protective role of fortified cookies in modulating oxidative-stress related conditions. As there are no standard products developed with the combination of these iconic pre-Hispanic foods as functional cookies or other interesting new technological uses, this research promotes Mexican renewable local bioresouces for a sustainable agri-food chain, especially edible mushrooms within the Sustainable Development Goals (SDGs).

## 5. Conclusions

This manuscript comprehensively illustrates that POF, NF and AF can be recommended as WWF fortifiers and applied as ingredients of functional products. The applicability of POF/NF/AF in production of novel fortified cookies was demonstrated. Partial replacement (50%) of WWF by biofunctional flours enhanced the nutritional value of cookies, particularly of those prepared with F2, in view of the obtained nutritional composition, the higher content of health promoting components and a significant antioxidant potential in comparison to control samples, without affecting sensorial properties. Consequently, cookies supplemented with POF/NF/AF flours had functional advantages over the conventional based WWF cookies, as well as good sanitary attributes and product acceptability for human consumption.

## Figures and Tables

**Figure 1 jof-07-00911-f001:**
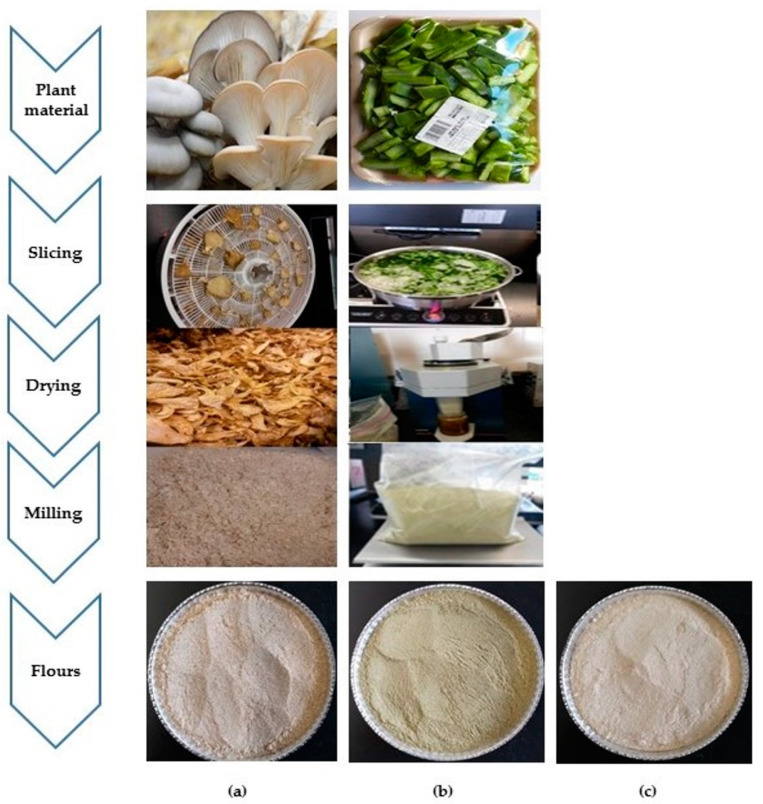
Plant material and its processing for biofunctional flour preparation. (**a**) *Pleurotus ostreatus* (oyster mushroom); (**b**) Nopal (*Opuntia ficus-indica* L.); (**c**) Amaranth (*Amaranthus* sp.). Photographs of commercial oyster mushrooms and nopal were taken from the companies’ websites (Hongos El Dorado, Mexico and COLPOS, Texcoco, Mexico, for oyster mushrooms and nopal, respectively). Photographs of powder making and flours were taken by Prof. Marcos Meneses–Mayo from Universidad Anáhuac Mexico.

**Figure 2 jof-07-00911-f002:**
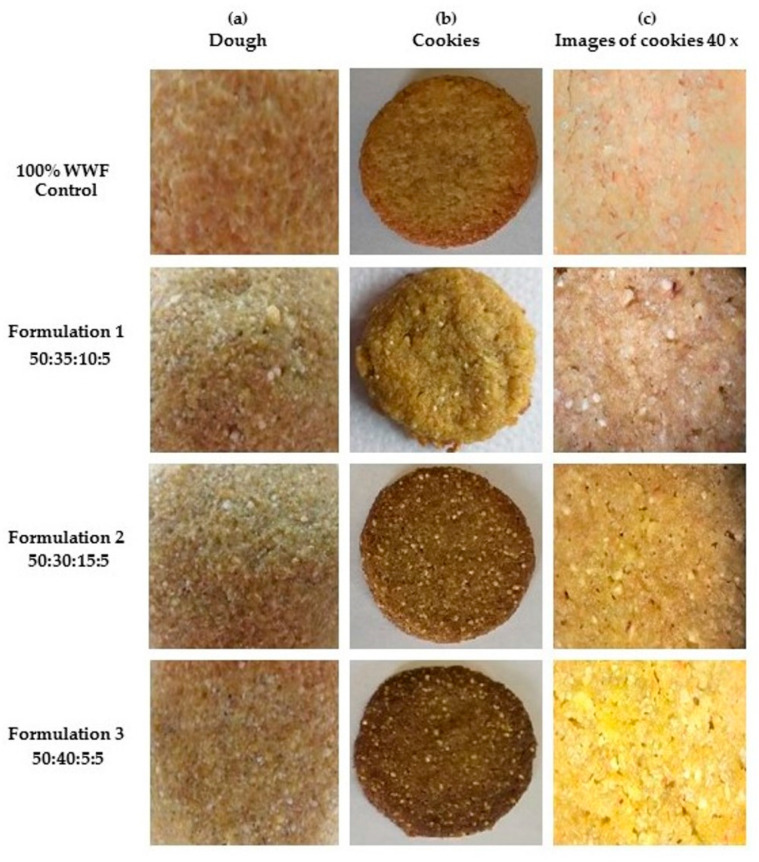
Representative photographs of doughs (**a**), obtained cookies (**b**), and the microscopical appearance of cookies at 40× magnification (**c**) for control and enriched cookies. The proportion of flours’ ingredients is indicated for each formulation as whole wheat:amaranth:nopal:*Pleurotus ostreatus* flours (WWF:AF:NF:POF).

**Table 1 jof-07-00911-t001:** Composition of control (whole wheat flour-based formulation) and those fortified with the functional flours used in cookies’ preparation (g/100 g of formulation).

Ingredients	Formulations			
	Control	F1	F2	F3
**Flours**				
Whole wheat flour (WWF)	34.9	17.4	17.4	17.4
*P. ostreatus* flour (POF)	-	1.7	1.7	1.7
Nopal flour (NF)	-	3.5	5.2	1.7
Amaranth flour (AF)	-	12.2	10.5	13.9
**Proportion of flours’ ingredients** **(WWF:AF:NF:POF)**	**100% WWF**	**50:35:10:5**	**50:30:15:5**	**50:40:5:5**
**Other ingredients**				
Margarine without salt	41.5	41.5	41.5	41.5
Sugar	19.9	19.9	19.9	19.9
Egg yolk	2.5	2.5	2.5	2.5
Vanilla extract	1.1	1.1	1.1	1.1
Salt	0.1	0.1	0.1	0.1

**Table 2 jof-07-00911-t002:** Proximal composition, energy value, phenolic and flavonoid contents, and antioxidant activity of WWF and the three alternative flours obtained from functional ingredients used in cookie formulations.

Parameter	WWF	POF	NF	AF
Moisture (%)	9.76 ± 0.03 ^a^	8.36 ± 0.29 ^b^	6.34 ± 0.21 ^c^	3.34 ± 0.45 ^d^
Water activity (a_w_)	0.38 ± 0.03 ^a^	0.37 ± 0.01 ^a^	0.34 ± 0.02 ^a^	0.16 ± 0.00 ^b^
Total protein (%)	16.37 ± 0.10 ^d^	23.96 ± 0.18 ^a^	18.99 ± 0.19 ^b^	18.21 ± 0.05 ^c^
Carbohydrates (%)	68.45 ± 1.42 ^a^	49.28 ± 0.49 ^b^	50.09 ± 0.62 ^b^	65.34 ± 1.02 ^a^
Total lipids (%)	2.07 ± 0.41 ^b^	1.04 ± 0.39 ^b^	1.16 ± 0.41 ^b^	7.26 ± 1.94 ^a^
Crude fiber (%)	1.99 ± 0.20 ^c^	10.37 ± 1.65 ^a^	6.95 ± 1.70 ^a^	3.18 ± 0.10 ^b^
Ash (%)	1.36 ± 0.02 ^d^	6.99 ± 0.17 ^b^	16.47 ± 0.14 ^a^	2.67 ± 0.10 ^c^
Energy value (kcal/100 g)	414 ± 3 ^c^	435 ± 2 ^b^	376 ± 0 ^d^	465 ± 0.00 ^a^
Total polyphenols (μg GAE/g)	516.3 ± 11.07 ^c^	599.3 ± 97.42 ^b,c^	4452 ± 46.50 ^a^	759 ± 17.71 ^b^
Total flavonoids (μg ECE/g)	11.25 ± 0.30 ^b^	12.52 ± 6.90 ^b^	64.72 ± 2.70 ^a^	9.97 ± 0.90 ^b^
Antioxidant activity (%, ABTS assay)	88.73 ± 3.11 ^b^	93.48 ± 2.61 ^a,b^	96.30 ± 0.10 ^a^	77.52 ± 1.74 ^c^

Different superscripts indicate significant differences of means for each parameter, according to a one-way ANOVA followed by a Tukey’s HSD (honestly significant difference) test (*p* < 0.05, *n* = 3).

**Table 3 jof-07-00911-t003:** Proximal composition, energy value, phenolic and flavonoid contents, and antioxidant activity of traditional WWF cookies and those produced with different formulations of alternative flours from functional ingredients.

Parameter	Control	F1	F2	F3
Moisture (%)	2.65 ± 0.15 ^b^	3.16 ± 0.15 ^a^	3.32 ± 0.04 ^a^	3.29 ± 0.01 ^a^
Water activity (a_w_)	0.41 ± 0.01 ^a,b^	0.44 ± 0.02 ^a^	0.33 ± 0.08 ^b,c^	0.25 ± 0.01 ^c^
Total protein (%)	7.07 ± 0.08 ^b^	8.17 ± 0.36 ^a^	8.18 ± 0.23 ^a^	7.93 ± 0.19 ^a^
Carbohydrates (%)	52.62 ± 1.24 ^a^	50.37 ± 0.48 ^b^	48.07 ± 0.66 ^c^	51.11 ± 1.52 ^a,b^
Total lipids (%)	34.10 ± 0.14 ^a^	33.32 ± 0.54 ^a,b^	33.13 ± 0.25 ^b^	33.58 ± 0.17 ^a,b^
Crude fiber (%)	2.58 ± 0.53 ^b^	3.11 ± 0.12 ^b^	5.29 ± 1.45 ^a^	2.54 ± 1.60 ^b^
Ash (%)	0.98 ± 0.03 ^c^	1.87 ± 0.05 ^a^	2.01 ± 0.07 ^a^	1.55 ± 0.06 ^b^
Energy value (kcal/100 g)	542 ± 37	586 ± 20	583 ± 10	590 ± 10
Total polyphenols (μg GAE/g)	272.1 ± 15.50 ^d^	1036 ± 42.07 ^b^	1366 ± 92.99 ^a^	677.6 ± 70.85 ^c^
Total flavonoids (μg ECE/g)	3.60 ± 2.70 ^b^	8.91 ± 1.80 ^a^	9.33 ± 1.20 ^a^	6.36 ± 3.00 ^a^
Antioxidant activity (%, ABTS assay)	75.60 ± 2.07 ^b^	89.58 ± 1.65 ^a^	88.82 ± 0.83 ^a^	87.73 ± 1.10 ^a^

Different superscripts indicate significant differences in means for each parameter, according to a one-way ANOVA followed by Tukey’s honestly significant difference (HSD) test (*p* < 0.05, *n* = 3).

**Table 4 jof-07-00911-t004:** Microbial counts of mesophilic microorganisms, yeasts and molds in traditional WWF cookies and those produced with different formulations of alternative flours from functional ingredients.

Determination (CFU/g)	Control	F1	F2	F3	Limits [37] (CFU/g)
Aerobic mesophilic microorganisms	<10	20	30	20	1 × 10^4^
Yeasts	<10	<10	<10	<10	300
Moulds	<10	<10	<10	<10	300

**Table 5 jof-07-00911-t005:** Sensory quality evaluation of traditional WWF cookies and those produced with different formulations of alternative flours from functional ingredients.

Sensory Attribute	Control	F1	F2	F3
Aroma	7.4 ± 0.97	7.3 ± 1.42	7.6 ± 0.70	7.6 ± 1.35
Color	7.1 ± 1.1	6.8 ± 1.55	7.5 ± 0.85	6.8 ± 0.92
Taste	7.3 ± 0.82	6.9 ± 1.10	7.8 ± 0.79	6.6 ± 1.71
Texture	6.8 ± 0.92	7.5 ± 1.27	8.0 ± 0.94	7.7 ± 0.95
Overall acceptability	7.3 ± 0.82	7.0 ± 1.41	8.0 ± 0.67	7.4 ± 0.84

*n* = 40 panelists. All data were expressed as the mean ± S.D. of sensory score of the aroma, color, taste, texture and overall acceptability of each cookie formulation. Data were analyzed according to a Kruskal–Wallis test followed by Dunn’s multiple comparison test. Statistical significance, *p* > 0.05.

## Data Availability

All data needed to evaluate the conclusion in the paper are present in the manuscript. Additional data related to this study are available on request to the corresponding authors.

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
