# Peer review of "Pre-Hispanic Foods Oyster Mushroom (Pleurotus ostreatus), Nopal (Opuntia ficus-indica) and Amaranth (Amaranthus sp.) as New Alternative Ingredients for Developing Functional Cookies"

_jof, 2021, doi:10.3390/jof7110911_

Round 1

Reviewer 1 Report

Abstract: There is no link between the section on NCDs and functional foods; it is recommended that information be restructured to try to make the wording more fluid.

Line 22: Correct “of flours´ ingredients”

Line 93: Beware of quotation marks - Recent trends in nutraceuticals´ research aroused scientists’-

Figure 1. It is recommended to improve the quality of the images by increasing the DPI's.

Line 266: What kind of "device"?

Line 295: What do you mean by "compared favorably", what are you trying to say? The sentence is not clear.

Page 9:  Line “Results on the physical characteristics of fortified cookies samples at all formulations showed similar weights”, In this sense, what did you expect to obtain? If masses of the same weight were formed, did you expect to obtain different weights?

Page 10: Line “When biofunctional flours were incorporated into cookies, a reduction of carbohydrates was observed coincidental with the increase in protein content” How can it increase the protein content? It is physically imposible increase the content. A decrease in protein content may occur due to the baking process, but never an increase.  This sentence should be modified as it does not correspond to the data in Table 3.

Page 10:  Section 3.3.2 Further discussion is suggested, for example no mention is made of the drastic decrease in polyphenol and flavonoid content after the baking process.

Page 10: Section 3.3.3. Further discussion of the results is suggested.

Table 3. To what is the decrease in fiber and ashes content in the formulated cookies attributed? It is recommended that this discussion be added to the section.

Discussion Section: Although there is sufficient comparison of the results found with others already published, it is recommended to try to explain and discuss more the results themselves, explaining the reasons for the findings. This part should be reinforced.

Conclusions: Remember that the conclusions section concludes with the objectives achieved in the project. The results data must there in the results section, it is not necessary to repeat this information. This section can be improved

Author Response

Dear reviewer! Many thanks for your valuable time and excellents comments to improve the quality of  our manuscript. We greatly appreciate your best efforts!!

Warmest regards from Cuba/Mexico

Reviewer 2 Report

Dear Authors

Good day!

I have read your manuscript, it is a well done job but yet I have two critical question and some comments.

Please fix comments as below:

Line 39. extra dash

Figure 1. If you can provide a higher resolution of photos that would be great or take new photos because they are pixel-ed, even alphabets. And if you are using downloaded photos from Internet you should indicate the source.

Line 168, 169, 190, 216, 222,223,... . There should not be any space between number and degree sign, or under lined! delete it here and in all repeats in the rest of manuscript. 

Line 179 ~ with regards to formulation; have you tried formulation for just WWF and others (POF, NF, and AF), just like 60:40 / 70:30? Please explain in this section what was your reason to do not include such combination.

Line 227. TPC stands for Total Phenolic Contents (not Compounds)

Line 231. "(75 g/L, 75 μL)" haven't you decided liquid or solid? which one is your final or if they are conveying different info please state them in the perantesis. 

Line 235. TF should be TFC; Total Flavonoid Contents

Line 248. You do not need to repeat the whole word again, continue with your abbreviation.

You can find my questions as follows: 

*For your information... I will finalize my decision to " Accept after minor revision" after receiving your answers to this two critical questions, so for now your manuscript is "major revision" grade for me.

  1. I had this general curiosity through whole results as how taste and aroma of different formulation differed. If they became more palatable or if they had distinct scent and taste. It could be done through questionnaire I assume. This is necessary for you to provide that info for your readers.
  2. With regards to the phrase "Prehispanic foods" in your tittle; why have not your consider MDPI-Foods journal? and as it is in manuscript tittle why you have not dedicated more to explain this food heritage? 

Regards;

Author Response

Dear editor! Many thanks for your valuable time and excellent comments for improving our manuscript. All my very best!!

Warmest regards

Reviewer 3 Report

Keywords section: the following list is suggested (amaranth, nopal, oyster mushroom, natural ingredients, functional foods, nutraceuticals, polyphenols, antioxidant, cookies)

Line 65: change the word "Particularly" by "In this context,"

Line 97: change "4000" by "4,000"

Line 116-127: according to the author's guide, only a brief mention of the main objective of the work should be made.

Line 139: Whole-wheat flour (WWF)

Line 141: …room temperature…

Line 146, 170, Table 1: …P. ostreatus

Line 146: Image quality needs to be improve. As a suggestion you could use a flow chart to facilitate the understanding of the sample processing.

Line 148: …at 57 °C (Grandmaster food dehydrator, Model FD-1010PC, Nesco® American Two Rivers, WI, USA). Afterward,…

Line 150: ….homogenized during 3 min (homogenizer FOSS 2094, Model XXX, Trademark??, State??, Conutry??)…

Line 151: …and then milled (food grinder Foss Tecator 1093, FOSS, State??, Mexico)…

Line 157, 159, 162, 165, 167, 175, 191: …Mexico…

Line 165: …were..

Line 170: POF and NF, change text format (without bold)

Line 171: ….State??, Italy)…

Line 174: AF, change text format (without bold)

Line 181: …WWF cookies (control)…

Line 185, 187: ..wheat flour or WWF?

Line 190: …for 12 min (convetion oven San-Son….

Line 193: magnification (microscope Olympus SZ4045, Olympus Corp., Tokyo, Japan).

Line 195: while or whole?

Line 216: …100 °C…

Line 217: change Bassersdorf by the abbreviate format

Line 223: ..4,000 rpm…

Line 235: …(TF) of extracts..

Line 239: …510 nm. The results were… (delete repeated information of the equipment)

Line 250: …at 734 nm. Ten …(delete repeated information of the equipment)

Line 252: …ABTS•+

Line 273: How many experimental repetitions with their respective replicas were carried out?

Line 284: …chemical composition of WWF and….

Line 283: no references should appear in results section

Line 301: ….exerted scavenging….

Line 302: continues numbering was missed from page 8 to 19; it is necessary to center the text of the cell contained in the table 2.

Page 8

-Table 2 (change ABTS by ABTS•+, modify through the manuscript)

Question: Why was ABTS only measured in the cookies, and was not commented with DPPH and reducing power methods?

Page 9:

-Paragraph 1: I don't understand how the physical tests (color) measured, was a sensory analysis performed to determine that one sample was darker than the other? or were measurements made with a colorimeter? How was this measurement carried out? Please indicate in the materials and methods section.

-First paragraph (no references should appear in results section)

-In figure 2, …delete under Olympus SZ4045 Stereo Microscope…due the equipment information should be appear in material and methods section.

Page 10

-Paragraph two (no references should appear in results section)

-Paragraph three: line 3 (especially in F2 and F1?); line 5(F 2 or F2); line 5 (delete 15% of nopal, the composition of this formulation is already indicated in the materials and methods section); line 7 …cookies (p > 0.05)…; line 9 …cookies samples (p > 0.05)…; line 10 …cookies (p < 0.05)…

-Paragraph four: line 1, 2, 4 and 10 (insert the abbreviated format of total phenolic and total flavonoid content, because they were previously abbreviated in the text); line 5 (…than F2, F1 and F3 had…); line 6 (…No significant differences (p > 0.05) were…; line 8 (…in F3 and F2, respectively.)

-Paragraph five: line 2 (no references should appear in results section)

-Paragraph six: line 2 (no references should appear in results section)

Page 11

-Table 3: homogenized the formulations like in table 1., p.e. F1, F2 and F3; traditional cookies or control like table 1?

Discussion section

-Paragraph 2: line 5 (development of FF/Ns appealing), the terms were abbreviated previously; line 6 (…enriched with POF, NF and AF is suitable…)

-Paragraph 4: line 2 (change nutraceuticals by Ns), due it was previously abbreviated; the table 2 should be cited only in results section; at the end section the discussion is compared with previous reports, but the references are missed

-Paragraph 5: line one (The P. ostreatus composition ranges….)

-Paragraph 6: line 1 (…Additionally, our results demonstrated that POF, NF and AF are vehicles….,such as polyphenols/flavonoids. These…); line 5 (explain the mechanism of the Folinc-Ciocalteu´s method); line 14 (….a direct relation (r2 = xxx??) between…)

Page 15

-Paragraph 1: line 5 (AF or AFs?)

-Paragraph 2: line 2 (105%??); line 3 (specially in F2 and F1 ?); line 4 (..fiber in F2…); this paragraph sounds more like a presentation of results section.

-Last paragraph: line 6 (..based on F2 represents…

Page 16

-Paragraph 2: last line (C. militaris)

-Paragraph 3: change F 2 by F2, as well as F 3 by F3; explain de mechanism of the ABTS method; change DPPH by DPPH through the manuscript.

Page 17

-Paragraph 2: Like the OH group and its number and distribution, what other structural factors play an important role in the antioxidant activity of phenolic compounds?

REFERENCES SECTION

Use ( – ) instead ( - ) in each reference page

Author Response

Dear editor!! Many thanks for your valuable time and excellent comments for improving our manuscript. We greatly appreciate that!!

Warmest regards

Round 2

Reviewer 2 Report

Dear Authors

Good day!

I have read the additional sensory section, It looked very good. 

one advice, I know it could get more expensive for your future works but in order to see the significant differences in sensory tests you have to increase your "n" probably to 100 or more but yet for this paper which the aim is JOF I guess that is enough. 

Regards

Reviewer 3 Report

The requested modifications were addressed correctly, so the manuscript can be considerd for publication